# Lung Ultrasound: A Useful Prognostic Tool in the Management of Bronchiolitis in the Emergency Department

**DOI:** 10.3390/jpm13121624

**Published:** 2023-11-21

**Authors:** Aiza C. Hernández-Villarroel, Alicia Ruiz-García, Carlos Manzanaro, Regina Echevarría-Zubero, Patricia Bote-Gascón, Isabel Gonzalez-Bertolin, Talía Sainz, Cristina Calvo, Mercedes Bueno-Campaña

**Affiliations:** 1Department of Paediatrics and Neonatology, Hospital Universitario Fundación Alcorcón, 28922 Madrid, Spain; aizacarh@ucm.es (A.C.H.-V.); aliruizgarcia@sescam.jccm.es (A.R.-G.); cmanzanaro@sescam.jccm.es (C.M.);; 2Paediatric Emergency Department, Hospital Universitario La Paz, 28046 Madrid, Spain; patricia.bote@salud.madrid.org (P.B.-G.); igbertolin@salud.madrid.org (I.G.-B.); 3Department of Paediatrics, Tropical and Infectious Diseases, Hospital Universitario La Paz, 28046 Madrid, Spain; talia.sainz@salud.madrid.org (T.S.); ccalvor@salud.madrid.org (C.C.); 4IdiPAZ Research Institute, Translational Research Network for Paediatric Infectious Diseases (RITIP), 28029 Madrid, Spain; 5Centre for Biomedical Research in Infectious Diseases (CIBERINFEC), 28029 Madrid, Spain; 6Department of Paediatrics, Autonomous University of Madrid, 28029 Madrid, Spain

**Keywords:** lung ultrasound, bronchiolitis, paediatric emergency medicine, prognosis

## Abstract

Lung ultrasound, a non-invasive bedside technique for assessing paediatric patients with acute respiratory diseases, is becoming increasingly widespread. The aim of this prospective, observational cohort study was to evaluate the effectiveness of a clinical ultrasound score in assessing infants with acute bronchiolitis in the emergency department and its ability to accurately identify patients at a higher risk of clinical deterioration. Infants under 6 months of age with clinical symptoms compatible with acute bronchiolitis were enrolled and underwent clinical and lung ultrasound evaluations. The study included 50 patients, the median age of which was 2.2 months (IQR: 1–5), and the primary outcome was respiratory support. Infants requiring invasive or non-invasive ventilation showed higher scores (5 points [IQR: 3.5–5.5] vs. 2.5 [IQR: 1.5–4]). The outcome had an AUC of 0.85 (95%CI: 0.7–0.98), with a sensitivity of 87%, specificity of 64%, and negative predictive value of 96.4% for a score <3.5 points. Children who scored ≥3.5 points were more likely to require respiratory support within the next 24 h (estimated event-free survival of 82.9% compared to 100%, log-rank test *p*-value = 0.02). The results suggest that integrating lung ultrasound findings into clinical scores when evaluating infants with acute bronchiolitis could be a promising tool for improving prognosis.

## 1. Introduction

Acute bronchiolitis is a significant health problem for infants and young children, as it stands among the primary causes of hospitalisation in infants under 12 months worldwide, particularly during autumn and winter seasons. It is defined as an acute viral infection affecting the lower respiratory tract, resulting in inflammation, oedema, mucus production, necrosis, and the subsequent obstruction of distal bronchioles, resulting in air being trapped in the lungs. It has the highest incidence and morbidity in infants aged under three months. Respiratory syncytial virus (RSV) is the most common pathogen.

At present, the condition’s diagnosis is mainly based on patients’ clinical histories and physical examinations. Routine laboratory tests are not recommended by current guidelines. In the absence of identifiable radiological patterns to aid diagnosis, imaging studies such as radiography are used primarily for the assessment of potential complications. Supportive care continues to be the primary approach to the management of the disease. This includes ensuring adequate fluid intake and providing respiratory support as needed, ranging from low-flow nasal cannula (LFNC) to mechanical ventilation (MV) [1,2].

Accurately predicting the severity of illness in infants presenting with acute bronchiolitis continues to be a major challenge. The overcrowding of paediatric wards and emergency departments during the autumn–winter seasonal outbreaks is a recurring issue that presents serious organisational and resource-related problems. With the aim of identifying patients at higher risk of clinical deterioration and therefore facilitating resource optimisation, several clinical scoring systems have been proposed to predict the severity of bronchiolitis [3,4,5,6]. However, these scoring systems are limited in their accuracy. 

In recent years, lung ultrasound (LUS) has become increasingly recognised as a valuable imaging diagnostic tool in assessing various paediatric and neonatal lung conditions. LUS boasts several advantageous features, including low-cost, relative safety, portability, and easy reproducibility, which minimises the avoidable exposure of children to ionising radiation. The basis of lung ultrasound lies in the analysis of artefacts resulting from changes in the acoustic impedance gradient. A healthy and adequately ventilated lung reflects ultrasound beams due to its high acoustic impedance. The ultrasound image displays a hyperechoic pleural line that exhibits a well-defined, regular, and dynamic movement during respiration (lung-sliding). This lineal image is followed by horizontal artefacts under the pleural line, called A-lines (horizontal, hyperechoic, static, and equidistant from each other), which are due to the reverberation of the ultrasound beam and indicate the normal presence of air. 

The usefulness of lung ultrasound in the management of bronchiolitis and in the assessment of patient prognosis is noteworthy and has been supported by numerous studies [7,8,9,10,11,12]. Although they are not specific, the lung ultrasound findings in bronchiolitis generally correspond to an irregular interstitial pattern with artefacts including B-lines and/or consolidations. These findings are related to the presence or absence of the previously mentioned pathophysiological changes, such as oedema, increased mucus production, and necrosis of the infected epithelial cells of the small airways, which result in a heterogeneous obstruction of the distal bronchioles [7]. Consequently, there is a decrease in the air content in the lungs, leading to impaired diffusion across the blood gas membrane and ventilation–perfusion incoherence. The severity of lung congestion can be assessed by measuring the concentration of vertical artifacts, which serve as a tool for quantifying pulmonary interstitial syndrome.

In 2019, a study was conducted by our group on hospitalised infants with acute bronchiolitis, resulting in the development of a predictive score (LUSCAB). This score is based on a combination of clinical and ultrasound variables [8]. To create the score, we performed a multivariate analysis of the influence of each ultrasound finding on the prognosis of needing respiratory support, along with traditionally associated severity variables such as age and WDF clinical score. The clinical ultrasound score was found to be an effective tool for identifying infants requiring respiratory support within the first 24 h of hospitalisation. However, the study was limited to patients with bronchiolitis who had already been admitted to hospital. We hypothesise that the LUSCAB score, which combines Lung UltraSound findings with Clinical data and Age in Bronchiolitis, could be used as a useful prognostic tool in the initial evaluation of patients presenting to the emergency department.

The aim of this study was to evaluate the accuracy of the LUSCAB score when used in the initial assessment of infants with acute bronchiolitis in the emergency department to rule out the potential need for respiratory support.

## 2. Materials and Methods

### 2.1. Study Population

A bicentric prospective observational cohort study was conducted in the paediatric emergency departments of two participating hospitals located in Madrid, Spain. Recruitment was conducted between November 2018 and January 2020, prior to the spread of the COVID-19 pandemic in Spain. The study protocol was approved by the local ethics committee and institutional review board of each participating centre. Written informed consent was obtained from the parents or legal guardians of all participants.

Infants under six months of age presenting in the emergency department with clinical symptoms consistent with acute bronchiolitis, according to McConachie’s definition [13], were eligible for inclusion. Exclusion criteria included chronic lung disease (bronchopulmonary dysplasia), cystic fibrosis, congenital heart disease with significant haemodynamic disorder, immunodeficiency, congenital or anatomical airway defects, and lack of parental consent. Patients were enrolled at the discretion of the attending paediatrician, with inclusion criteria restricted to the first presentation to the emergency department with the specified symptoms.

A standardised data sheet was used to collect patient demographic, medical, clinical, and ecographical data. Patients underwent a comprehensive clinical evaluation, including a thorough investigation of their medical history, and were managed according to the standard clinical protocol at each participating centre, including requests for additional tests, recommendations for hospital admission, and the initiation of respiratory support when appropriate. Upon arrival, vital signs were recorded, including respiratory rate and oxygen saturation levels in ambient air. The modified Wood–Downes–Ferres (WDF) score (Table 1), commonly used in the two participating centres, was employed for the clinical assessment of severity. Admission was primarily indicated for patients with symptoms of moderate to severe bronchiolitis (WDF score greater than 4), hypoxaemia (SaO_2_ less than or equal to 92% awake or less than or equal to 90% asleep), presence of apnoea, need for intravenous rehydration, parental concern, or any other social difficulties. 

Patients were monitored from the time they arrive at the emergency department to discharge (either from the emergency department or the ward, depending on the case). In addition, any re-consultations in the emergency department after discharge and whether the patient required admission were also documented. Respiratory viruses were identified only in admitted patients by antigen and/or PCR testing, depending on hospital availability. Testing was conducted through immunochromatography (Standard Diagnostics™ Suwon, Republic of Korea) or a polymerase chain reaction (Alere i RSV™, Abbott, Chicago, IL, USA). Respiratory support and oxygen therapy were delivered via a variety of methods, ranging from low-flow nasal cannula to high-flow nasal cannula, as well as non-invasive (NIV) and invasive mechanical ventilation (MV). For the purpose of data analysis, the study examined the use of high-flow nasal cannula (HFNC), nasal continuous positive airway pressure (nCPAP), and bilevel positive airway pressure (nBiPAP) as forms of non-invasive ventilation.

After the attending paediatrician made the medical decision to discharge or admit the patient to hospital, a paediatrician from the research team performed a lung ultrasound according to the pre-agreed methodology (see Section 2.2). The lung ultrasound findings did not have any impact on the medical decisions. The duration of the lung ultrasound procedure was recorded for every patient. A research team member, also responsible for recording clinical outcome data, applied the LUSCAB score to all enrolled patients in a subsequent phase. The LUSCAB score, as presented in Table 2, combines the ultrasound findings (such as the presence and location of B-lines, confluence of B-lines, and/or consolidation) with the Wood–Downes–Ferres score and epidemiological data (age) [8].

### 2.2. Lung Ultrasound Methodology

The research team consisted of six paediatricians with proven and documented experience in the performance of clinical lung ultrasound, a technique that has demonstrated reliable interobserver agreement in previous studies [7,15,16]. All collaborating clinicians participated in a training session to ensure a common systematic approach to examination and data collection prior to patient recruitment. Ultrasound studies were conducted according to the standard methodology described by Copetti et al. [17]. The imaging protocol involves obtaining longitudinal sections by placing the probe in a longitudinal position on the anterior chest wall, along the anterior parasternal and medio-axillary lines bilaterally, and on the posterior thorax, along the paravertebral lines bilaterally. Probe displacement is performed longitudinally from cephalad to caudal, evaluating each intercostal space. The ultrasound equipment used included a LOGIQ V2 ultrasound machine (General Electric International TM, Madrid, Spain), which was equipped with a 5–13 MHz linear probe and a Mindray DC-40 ultrasound machine (Mindray Medical España S.L., Madrid, Spain), which was equipped with a 6–14 MHz linear probe. 

Longitudinal and transversal sections of the anterior, lateral, and posterior chest wall were performed. Ultrasound findings were recorded for the anterior and posterior areas using a standardised data sheet with a thoracic diagram, which was pre-designed in our previous study [8]. Pathological ultrasounds findings to be evaluated include: the presence of more than three B-lines per intercostal space (hyperechogenic vertical lines perpendicular to the pleural line from the bottom of the screen that remain distinguishable from each other), confluent B-lines (unilateral or bilateral), and subpleural consolidations (echo-poor subpleural region interrupting the pleural line, measuring more or less than 1 cm). The normal pattern is defined as a normal lung sliding with A-lines and less than three B-lines per intercostal space. Short vertical artefacts and isolated B-lines are considered clinically non-significant findings [7,8,10].

### 2.3. Statistical Analysis

The study presents quantitative data as mean and standard deviation or median and interquartile range, and qualitative data as counts and percentages. Univariate analysis was used to compare groups defined by respiratory support type and emergency department clinical decision. For qualitative variables, the Chi-square test or Fisher’s exact test was performed. For quantitative variables, the non-parametric Kruskall–Wallis test was used. 

The discriminatory ability of the predictive score for any form of respiratory support, need for oxygen therapy, and admission was evaluated by calculating the area under the receiver operating characteristic (ROC) curve. Time to respiratory support within groups determined by the predictive score cut-off point (≥3.5 points) was analysed using Kaplan–Meier survival curves. The log-rank test was used to compare the corresponding curves. All tests underwent bilateral evaluation, with *p* values being considered statistically significant at *p* < 0.05. Data analysis and test performance estimations were conducted using commercially available software (SPSS version 16.0 for Windows, SPSS Inc., Chicago, IL, USA and STATA 12).

## 3. Results

A total of 50 patients with a median age of 2.2 months (interquartile range 1–5 months) were enrolled, of whom 60% were male. The most common virus implicated was RSV, accounting for 62% of cases. The median time between the onset of symptoms and the presentation to the emergency department was four days (interquartile range 3 to 5 days), and the median WDF score upon arrival was 4.5 (interquartile range 3–6). The median duration for performing lung ultrasound (LUS) was eight minutes with a standard deviation (SD) of two. Table 3 provides a summary of the clinical and epidemiological data of the participants. 

Twenty-six patients were discharged following their first visit to the emergency department. Seventeen patients revisited the emergency department within 24 h, and nine were subsequently admitted to the hospital following re-consultation. Patients who were discharged after their initial emergency department visit and did not require admission even after eventual re-consultation were found to be older (3.7 [SD: 1.9] vs. 1.9 [SD: 1.7] months, *p* = 0.001) and had a lower LUSCAB score (2.5 [IQR: 1.3–3.2] vs. 3.7 [IQR: 2.5–5.0] points, *p* = 0.016) compared to those patients who were eventually admitted. No significant differences were found in regard to other variables such as WDF score, room air oxygen saturation levels on arrival, and time from symptom onset to lung ultrasound. 

Out of the complete cohort of 50 patients, 33 (66%) were eventually admitted to the hospital, and 21 (42%) required respiratory support. Of the 21 patients, only 3 received low-flow nasal cannula exclusively, whereas the rest required supplementary respiratory support in tandem with the clinical course: 10 patients were administered high-flow nasal cannula, 7 required non-invasive ventilation, and only 1 needed invasive mechanical ventilation. No correlation was observed between oxygen saturation levels during emergency department admission and the LUSCAB score (ρ = −0.89, *p* = 0.5).

Patients who did not require any respiratory support had a significantly lower LUSCAB score compared to patients who required non-invasive or invasive ventilation, including nCPAP/nBiPAP/MV (2.5 [IQR: 1.5–4] versus 5 points [IQR: 3.5–5.5] *p* = 0.006). Table 4 presents the clinical and epidemiological data of patients according to their need for respiratory support. 

The LUSCAB score shows acceptable discriminatory ability in the assessment of need for respiratory support (excluding low-flow nasal cannula), with an area under the curve (AUC) of 0.75 (95% CI: 0.6–0.89). However, its discriminatory capacity is higher for the most precise requirement of nCPAP/nBiPAP/MV, having an AUC of 0.85 (95% CI: 0.71–0.98). A LUSCAB score of less than 3.5 points shows a sensitivity of 87% and a specificity of 64% to exclude the specific need for nCPAP/nBiPAP/MV, yielding a negative predictive value of 96.4%. The AUC values, sensitivity, specificity, as well as negative and positive predictive values for the LUSCAB score concerning the need for any type of respiratory support or admission are presented in Table 5. 

Nineteen patients who achieved a LUSCAB score of 3.5 or higher were more susceptible to requiring any type of respiratory support within the subsequent 24 h of emergency department evaluation (with an estimated event-free survival rate of 54.1% compared to 76.1%, log-rank test *p* = 0.06). Additionally, they were more likely to require nCPAP/nBiPAP/MV specifically (with an estimated event-free survival rate of 82.9% compared to 100%, log-rank test *p* = 0.02) compared to patients with a LUSCAB score of less than 3.5. 

## 4. Discussion

Despite its small sample size, the preliminary results of this study suggest that the LUSCAB score, which combines clinical and ultrasound variables, may have value in the assessment of infants with acute bronchiolitis presenting to emergency departments. Prior studies have explored the potential use of lung ultrasound in predicting the necessity for respiratory support [15,18], hospital admission [12], requirement for oxygen therapy [7,15,18], and length of hospital stay [15]. However, this study is among the first to integrate lung ultrasound findings with clinical and epidemiologic data, thereby increasing the accuracy of ultrasound findings for the evaluation of patients with acute bronchiolitis in the emergency department.

Our findings show that children with acute bronchiolitis who eventually required respiratory support had a higher LUSCAB score during their initial emergency department evaluation. The LUSCAB score was found to be a reliable indicator for the requirement of respiratory support with nCPAP/nBiPAP/MV, exhibiting good discriminatory ability (AUC 0.85, 95% CI: 0.72–0.98). Implementing a threshold of 3.5 points on the LUSCAB score demonstrated high sensitivity in ruling out the necessity for nCPAP/nBiPAP/MV, with a negative predictive value of 96.4%. The results obtained support the hypothesis that the identification of lung aeration impairment can be facilitated by the presence of pathological findings on ultrasound, such as those included in the LUSCAB score. Thus, this could be useful in predicting clinical outcomes and as a marker of the severity of the disease [19]. 

If validated in larger studies, these findings would provide evidence for the utility of the LUSCAB score in identifying patients at higher risk of worse clinical outcomes. In other words, incorporating lung ultrasound in the assessment of patients with bronchiolitis could enhance their evaluation and increase confidence in safely discharging patients who are considered to be at a lower risk. The early identification of high-risk patients would significantly contribute to improved hospital management, facilitate decision making regarding patient admission, and allow for the timely and accurate administration of respiratory supportive treatment [20,21].

Interestingly, despite its accuracy in identifying patients who may require respiratory support, the LUSCAB score demonstrates limited capability in predicting admission. This is potentially due to the multifaceted nature of admission criteria, which are occasionally not directly linked to the severity of the illness, such as age, feeding problems, and social difficulties like parental anxieties.

Furthermore, the ability of the LUSCAB score to predict the requirement for oxygen therapy has also not been demonstrated to be good. The median oxygen saturation level on arrival was 96% (IQR: 94–98%), and there was no discernible difference between those who finally required oxygen therapy and those who did not. In addition, no correlation was found between LUSCAB score and room air oxygen saturation levels when performing lung ultrasound. These findings are consistent with previous published studies that suggest that the target oxygen saturation level may determine admission rates and/or length of stay without affecting the rate of respiratory support intensification [7,8,11,12,15,18].

The results of the present study showed that the median WDF score on arrival was 4.5 (IQR: 3–6), indicating that more than 50% of participants experienced moderate to severe respiratory distress. Nonetheless, there was no discernible difference in the WDF score between those who eventually required some kind of respiratory support and those who did not. As mentioned above, those who eventually required respiratory support had a significantly higher LUSCAB score, which supports the theory that combining lung ultrasound findings with clinical data (WDF score) improves the discriminative ability of the WDF score alone, as previously described in other studies [8,15].

One limitation of this exploratory study is undoubtedly its small sample size, brought about by the inclusion criterion of less than 6 months of age, which aimed to reduce the likelihood of including other pathologies that may mimic bronchiolitis [22,23]. Our ability to generalise the findings to older children may therefore be limited. 

It should also be noted that the majority of patients presented to the emergency department four days after symptom onset, which could potentially compromise our ability to identify high-risk patients at an earlier stage of disease. Additionally, the high rate of hospital admissions poses another limitation, as cases of mild or non-severe acute bronchiolitis may be underestimated.

### Application and Recommendations

Point-of-Care Ultrasound (POCUS) is increasingly being used for diagnostic purposes in a variety of healthcare settings due to its rapidity, portability, repeatability, and non-ionising properties. Specifically, lung ultrasound (LUS) is proving to be a valuable tool for the objective assessment of respiratory disease in adults, children, and neonates [17,19,20]. Although it is considered less convenient than radiology due to the time required to perform it and the need for a learning curve, this technique is, in fact, a relatively simple and easy-to-learn procedure, with a median time to perform and interpret lung ultrasound findings of only 6 min [16]. The identification of pathological findings is not overly challenging but requires a systematic approach and interpretation based on the clinical context, since neither lesions nor ultrasound findings are exclusively indicative [24,25]. Since the onset of the COVID-19 pandemic, there has been an exponential increase in the use of this imaging diagnostic test [26]. This may be attributed to the convenience of performing the test at the patient’s bedside, which reduces the need for transfers and provides opportunities for repeat testing throughout the course of the disease. This facilitates comprehensive pathogenic progression monitoring, empowering the creation of customised treatment plans for each individual patient. With the increasing availability of new portable devices, point-of-care lung ultrasound is becoming more widely used in emergency departments, where clinical examination and this technique may complement each other in the assessment of patients with acute respiratory distress, making it an encouraging tool for clinical practice [21].

## 5. Conclusions

In summary, our preliminary results suggest that the LUSCAB score, which combines lung ultrasound findings with clinical and epidemiological data, may be a valuable prognostic tool for identifying patients with acute bronchiolitis at increased risk of requiring respiratory support within 24 h of initial emergency department assessment. Nonetheless, further extensive studies are needed to confirm the usefulness of the LUSCAB score in this setting. If validated, this scoring system could serve as a suitable tool for screening and identifying infants at a higher risk of clinical deterioration. This would aid emergency clinicians in safely discharging infants with acute bronchiolitis.

## Figures and Tables

**Table 1 jpm-13-01624-t001:** Wood–Downes clinical scoring system modified by Ferres [14].

	0	1	2	3
Wheezing	None	End expiration	Entire expiratory phase	Inspiration and expiration
Retractions	None	Subcostal or lower intercostal	1 + supraclavicular + nasal flaring	2 + suprasternal + lower intercostal
Respiratory rate-breaths/min	<30	31–45	46–60	>60
Heart rate-beats/min	<120	>120		
Inspiratory breath sounds	Normal	Regular, symmetrical	Markedly silent, symmetrical	Silent thorax, no wheezing
Cyanosis	Not present	Present		

A score of 1–3 points denotes mild bronchiolitis; 4–7 moderate bronchiolitis; and 8–14 severe bronchiolitis.

**Table 2 jpm-13-01624-t002:** LUSCAB (Lung Ultrasound findings and Clinical data and Age in Bronchiolitis) score.

LUSCAB Score	Points
Age < 1 month	1.5
LUS ^a^ findings	
>3 B lines per ICS ^b^ bilateral (anterior area)	1.5
Confluents B line bilateral (anterior area)	1
One or more posterior consolidations <1 cm	1
One or more posterior consolidations >1 cm	3
WDF ^c^ score ≥ 6 points	2.5

^a^ Lung Ultrasound; ^b^ Intercostal Space; ^c^ Wood–Downes–Ferres.

**Table 3 jpm-13-01624-t003:** Clinical and epidemiological data.

	Patients (*n* = 50)
Age, months (IQR)	2.18 (1.06–4.86)
Age < 1 month, *n* (%)	11 (22%)
Sex, male, *n* (%)	30 (60%)
Aetiology	
Unknown, *n* (%)	18 (36%)
RSV ^a^ infection, *n* (%)	31 (62%)
Influenza virus, *n* (%)	1 (2%)
Time from onset to ED ^b^ assessment, days (IQR)	4 (3–5)
WDF ^c^ score (IQR)	4.5 (3–6)
LUSCAB score (IQR)	2.5 (1.5–4)
Temperature >37.5 °C, *n* (%)	24 (48%)
Mean room air SaO_2_ in ED ^b^, % (SD)	95.7 (2.7)
Hospital admission, *n* (%)	33 (66%)
Oxygen therapy	3 (6%)
Respiratory support, *n* (%)	18 (36%)
Only HFNC ^d^, *n* (%)	10 (20%)
nCPAP ^e^/nBiPAP ^f^, *n* (%)	7 (14%)
Mechanical Ventilation, *n* (%)	1 (2%)
PICU ^g^ admission	1 (2%)
Mean length of stay, days (IQR)	4 (3–6)
Death, *n* (%)	0 (0%)

^a^ Respiratory Syncytial Virus; ^b^ Emergency Department; ^c^ Wood–Downes–Ferres; ^d^ High-Flow Nasal Cannula; ^e^ nasal Continuous Positive Airway Pressure; ^f^ Bilevel Positive Airway Pressure; ^g^ Paediatric Intensive Care Unit.

**Table 4 jpm-13-01624-t004:** Clinical and epidemiological data according to need for respiratory support.

	No Support *n* = 29	LFNC ^d^/HFNC ^e^ *n* = 13	nCPAP ^f^/nBiPAP ^g^/MV ^h^ *n* = 8	*p* Value
Sex male, *n* (%)	18 (62)	7 (54)	5 (62)	ns
Age < 1 month, *n* (%)	3 (10)	4 (31)	4 (50)	ns
SaO_2_ on arrival, mean (SD)	96.3 (2.4)	95.1 (2.2)	94.4 (4.1)	ns
Onset to ED ^a^ (h), median (IQR)	4 (2–5)	4 (3–6)	4 (3–6.5)	ns
Admission/respiratory support, *n* (%)	12 (41)	13 (100)	8 (100)	<0.001
LOS ^b^ (days), median (IQR)	3 (2–3.75)	4 (3–6)	8 (5.3–11.8)	<0.001
WDFs ^c^ on arrival, median (IQR)	4 (3–5.5)	5 (3.5–6)	6 (3–9.25)	ns
LUSCAB median (IQR)	2.5 (1.5–4)	2.5 (1.3–4)	5 (3.5–5.5)	0.006

^a^ Emergency Department; ^b^ Length of Stay; ^c^ Wood–Downes–Ferres score; ^d^ Low-Flow Nasal Cannula; ^e^ High-Flow Nasal Cannula; ^f^ nasal Continuous Positive Airway Pressure; ^g^ nasal Bilevel Positive Airway Pressure; ^h^ Mechanical Ventilation; ns, not significant.

**Table 5 jpm-13-01624-t005:** AUC, sensitivity, specificity, and predictive values of LUSCAB score less than 3.5 points for different outcomes.

	AUC ^a^ (CI95%)	Se ^b^	Sp ^c^	PPV ^d^	NPV ^e^
Admission	0.64 (0.47–0.80)	51.5%	70.6%	77.3%	42.9%
Oxygen therapy	0.69 (0.53–0.84)	61.9%	69.0%	59.0%	71.4%
Respiratory support	0.75 (0.60–0.89)	66.7%	68.8%	54.6%	78.6%
nCPAP ^f^/BiPAP ^g^/MV ^h^	0.85 (0.72–0.98)	87.5%	64.3%	31.8%	96.4%

^a^ Area under the curve; ^b^ Sensitivity; ^c^ Specificity; ^d^ Positive predictive value; ^e^ Negative predictive value; ^f^ nasal Continuous Positive Airway Pressure; ^g^ Bilevel Positive Airway Pressure; ^h^ Mechanical Ventilation.

## Data Availability

The data presented in this study are available from the corresponding author upon request.

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
