# Peer review of "Lung Ultrasound: A Useful Prognostic Tool in the Management of Bronchiolitis in the Emergency Department"

_jpm, 2023, doi:10.3390/jpm13121624_

Round 1
Reviewer 1 Report
Comments and Suggestions for Authors
POCUS, or point-of-care ultrasound, is a promising imaging test for pediatric respiratory infections. Its use in evaluating infants with bronchiolitis has significant clinical value. The Hernández-Villarroel et al. study is a bicentric, real-life perspective study that assesses the combined score, LUSCAB, which includes both clinical and imaging data. This is a great advantage of the study. The research is well-conducted, and the manuscript accurately represents the study's aim, design, materials, and methods. The results are well-described and sufficient, and the discussion is profound and informative. However, I suggest including an additional subsection on practical implementation and recommendations.
Author Response
Dear reviewer, thank you for your positive feedback. In accordance with your suggestion, an extra section with recommendations for the practical use of lung ultrasound in relation to the current evidence has been added at the end of the discussion. We hope that these changes will meet your requirements. Best regards.
Reviewer 2 Report
Comments and Suggestions for Authors
This study reports the use of a score called LUSCAB, that combines lung ultrasound findings, age and clinical data to predict outcome in infants examined in the emergency department for bronchiolitis. According to the authors, LUSCAB may be a promising tool that may help the pediatrician to identify high-risk patients, requiring respiratory support during the course of the disease.
This is an interesting manuscript, that, although the small number of cases, could add value to current knowledge. However, there are several points that should be addressed to strengthen the quality of the study:
- lines 14-16: the sentence is to long and should be simplified. "Aims of the study were...."
- line 19 : "key concern" is not the appropriate term. Do the authors mean primary endpoint/outcome?
- line 25: The sentence is not correct. Modify as follows: "incorporating lung ultrasound findings"
- line 26: "enahance the prognosis" does not male sense.
- line 43: please modify "restricted" in "limited"
- Table 3: please specify which of the difference between the three groups was statistically significant: the first, the second or both?
- Kaplan-meier curves are mentioned in the methods section but not presented in the results section.. Please calrify.
Comments on the Quality of English LanguageModerate english editing is required prior to publication.
Author Response
Dear reviewer, thank you for your positive feedback.
Following your recommendations, the following changes have been made:
- Lines 14-16 have been simplified to help clarify.
- line 19: this refers to the primary outcome.
- line 25: the sentence has been changed to "incorporating lung ultrasound findings"
- line 26 has been reformulated.
- The term "limited" is now used instead of "restricted" in line 43.
- Table 3: there is a statistically significant difference in outcomes between patients who did not receive ventilation (first group) and those who required nCPAP/nBiPAP/MV (third group) (2.5 [IQR: 1.5-4] vs. 5 points [IQR: 3.5-5.5], p= 0.006). Ultrasound was found to have a better predictive value for patients who received this type of respiratory support.
- The Kaplan-Meier curves are analysed in the results section, with approximate survival rates and log-rank test evaluation of the curves. The analysis graphs are available.
We hope that these changes will meet your requirements.
Best regards.